# Prehabilitation Prior to Chemotherapy in Humans: A Review of Current Evidence and Future Directions

**DOI:** 10.3390/cancers17162670

**Published:** 2025-08-15

**Authors:** Karolina Pietrakiewicz, Rafał Stec, Jacek Sobocki

**Affiliations:** 1Faculty of Health Sciences, Medical University of Warsaw, 02-091 Warsaw, Poland; 2Department of Oncology, Medical University of Warsaw, 02-091 Warsaw, Poland; 3Department of General Surgery and Clinical Nutrition, Centre of Postgraduate Medical Education, 00-416 Warsaw, Poland

**Keywords:** cancer, chemotherapy, prehabilitation, integrative review, QOL

## Abstract

Chemotherapy presents significant challenges for patients, as it often causes unpleasant side effects. It also creates considerable strain on medical staff, who must respond promptly to complications in order to help patients complete treatment in the best possible condition. The interval between diagnosis and the start of therapy, typically several weeks long, represents an invaluable window during which registered healthcare professionals can implement prehabilitation interventions—nutritional support, physical activity programmes, psychological counselling, and assistance with smoking cessation. These measures have great potential to elevate the patient’s baseline functional status, improve treatment outcomes, and increase both survival rates and quality of life. Given the significant gaps in the literature on this topic, we conducted an integrative review of existing studies in this field. Our findings provide the foundation that may guide future researchers in designing clinical trials and developing effective prehabilitation protocols.

## 1. Introduction

The usual social behaviour related to nutrition, physical activity, and general health care remains far from satisfactory, contributing to a progressive decline in overall well-being, even among otherwise healthy individuals [1]. Chronic diseases and cancer significantly reduce the ability to preserve health and functional status. Good physical fitness and nutrition increase the likelihood of tolerating chemotherapy with fewer or no adverse effects and completing the full treatment regimen without dose reductions. Chemotherapeutics used for oncological treatment not only kill cancer cells, with the aim of curing the disease, but also cause a variety of side effects that influence patient quality of life (QOL). Side effects most frequently include fatigue, loss of appetite, nausea, vomiting, diarrhea, dyspnoea, psychological distress, reduced cardiac function, early menopause, or peripheral neuropathy [2,3,4,5,6,7].

Time from diagnosis to treatment (TDT) or time to treatment initiation (TTI) can differ depending on several factors, including the type of cancer, the patient’s condition, healthcare system efficiency, and access to medical care; therefore, defining an average TDT seems unfeasible. A retrospective cohort study by Souza [8] of patients with diffuse large B-cell lymphoma showed a median TDT of 19 days, whereas in a cohort study (*n* = 2,241,706) by Cone [9], the median TTI for prostate cancer was 79 days, 26 days for colon cancer, 32 days for breast cancer, and 41 days for non-small-cell lung cancer.

Prehabilitation is defined as a single or set of medical and health interventions that mainly pertain to nutrition, physical activity, counselling, or smoking cessation, beginning at the point of diagnosis and lasting until the initiation of acute treatment [10,11]. It aims to enhance or preserve an individual’s psychological and physiologic reserves, which are predicted to decline due to cancer therapy. During that process, medical staff are assumed to define the individual’s baseline functional levels and determine focused interventions that would minimize negative effects of the therapy and improve patient QOL while simultaneously improving treatment outcome [12]. Currently, in the literature the main focus is on prehabilitation prior to surgery, but seldom prior to radiotherapy or chemoradiation, although there is immense paucity of papers exploring its efficacy prior to chemotherapy [13]. Perioperative care has been developing since 1990s, when it was promoted by the Enhanced Recovery After Surgery (ERAS) society. This organization provides guidelines on perioperative care before cancer surgery, as well as before major surgeries not connected with oncology [14].

These observations highlight the need for prehabilitation protocols to optimize patients’ overall health—encompassing physical fitness, e.g., VO_2_max (maximal oxygen uptake), nutritional reserves, mental well-being, and addiction cessation—before initiating chemotherapy treatment. Establishing such protocols is essential, as baseline nutritional status, physical condition, and psychological health indisputably influence treatment tolerance, long-term prognosis, and postoperative outcomes. Therefore, the intent of this review is two-fold. Initially, we aim to synthesize knowledge about examined methods used to improve patient’s health status before undergo chemotherapy. Additionally, we aim to determine whether these findings allow for the delineation of a protocol of prehabilitation for chemotherapy.

Accordingly, the aim of this integrative review is to assemble the currently available data on prehabilitation interventions applied to cancer patients prior to chemotherapy and to identify what is relevant and what is missing in the literature to formulate the protocol. To date, to the best of our knowledge, no review has been published that summarizes the existing evidence on medical support before chemotherapy.

Our review reveals a limited number of studies on prehabilitation prior to chemotherapy. However, the available evidence highlights the critical role that interventions implemented between diagnosis and treatment initiation may play in improving therapy success, reducing adverse effects, and enhancing survival rates. There is a clear need for further experimental research, particularly focusing on protocols involving physical activity interventions, nutritional support, and counselling, to better understand their effectiveness and optimize prehabilitation strategies.

## 2. Materials and Methods

Given the importance of the topic, we conducted an integrative review guided by Torraco [15] and originally conceptualized by Whittemore [16]. This method focuses on assembling and synthesizing both experimental and non-experimental studies to summarize existing knowledge and generate new perspectives. We selected the integrative review because of the limited literature and small number of studies available on our chosen topic. We conducted an analysis of available papers on 13 January 2025, searching the MEDLINE/PubMed (Medical Literature Analysis and Retrieval System Online/National Library of Medicine), Cochrane, Scopus, and Web of Science databases, in order to identify research concerning prehabilitation prior to chemotherapy published between 2010 and 13 January 2025 in English. Firstly, we made a list consisting of all keywords related to the topic of this review. Afterwards, the thesauruses of the four databases were consulted to determine which keywords they use to locate literature. As a result, we identified four keywords for our final search—“chemotherapy”, “drug therapy”, “neoadjuvant”, and “prehabilitation.” In the Scopus search, the search query included drug AND therapy instead of the exact phrase "drug therapy", which may have retrieved records where the words appeared independently. However, all included studies were manually screened for relevance to drug therapy, reducing the risk of including irrelevant results. Due to the limited number of studies on prehabilitation in chemotherapy settings, we decided to include studies focusing on patients prior to adjuvant chemotherapy. The initial search yielded 162 articles (MEDLINE/PubMed *n* = 31, Cochrane *n* = 15, Scopus *n* = 44, Web of Science *n* = 72). An overview of this search stage is presented in Table 1. The identified articles were imported into EndNote21 for further analysis.

The extracted research articles had to involve human subjects aged 18 and above and concern oncological patients prior to chemotherapy. Both non-experimental and experimental studies were included. We focused exclusively on chemotherapy-related papers, consequently excluding those addressing surgery, radiotherapy, chemoradiotherapy, hormone therapy, or immunotherapy. Furthermore, studies concerning ERP (Enhanced Recovery Therapy) and ERAS (Enhanced Recovery After Surgery) protocols were rejected, as both entail pre-, intra-, and post-operative management. Only studies published in peer-reviewed journals were considered eligible for inclusion. All inclusion and exclusion criteria are summarized in Table 2.

Initially, duplicates were removed (*n* = 42) using EndNote21 reference management software, leaving 120 articles. A further 98 papers were then excluded based on title and abstract screening, using our predefined criteria (Table 2). We did not employ a separate appraisal checklist or software package; instead, the criteria themselves comprised our screening tool. Twenty-two studies were retained for full-text review. Upon comprehensive evaluation against the established inclusion and exclusion criteria, 12 papers met the requirements.

To ensure reliability, all authors independently screened the same 22 full-text articles, and any disagreements were resolved by consensus. Two potentially eligible book chapters (Watson [17], Brown [18]) were excluded from the final synthesis due to the unavailability of their full texts despite attempts to retrieve it. Although their abstracts suggested relevance, full assessment was not possible. The screening strategy is illustrated in Figure 1.

A final set of 10 papers were obtained and included in the review. All retrieved articles categorized by study design, type of prehabilitation, study structure, and cancer type are presented in Table 3.

The majority of the included studies were reviews focusing on nutritional, physical, and psychological (counselling) interventions, as illustrated in Figure 2.

## 3. Results

### 3.1. Nutritional Interventions

Assessment tools are commonly used by scientists in their research and one of them is muscle loss screening. Its importance in preventing sarcopenia, which is a critical factor for survival rate and mortality among oncological patients, was emphasized in the paper by Jang [24]. Among others, strong associations between low muscle mass and poor survival in cancer patients have been demonstrated in the TRACERx (Tracking Non-Small-Cell Lung Cancer Evolution through Therapy) lung study [28] as well as in existing systematic reviews, as referenced by Phillips [29]. In the context of nutrition, Mina [11] in their review points out that adequate protein intake is essential in preventing sarcopenia, which raises breast cancer (BC) treatment toxicity and advances tumour progression. Eating between 1.2 and 2.0 g/kg of protein may prevent sarcopenia in patients with cancer [30]. These findings further confirm that sarcopenia is a significant issue among cancer patients, and that recommendations regarding prehabilitation for patients awaiting chemotherapy should be established. Besides sarcopenia, another problem that may affect cancer treatment is bone loss among elderly patients (≥65 years old) in whom the use of certain chemotherapeutic agents may lead to cancer treatment-induced bone loss [31,32,33]. Bertrand [23] indicates that screening for vitamin D deficiency and calcium levels during Comprehensive Geriatric Assessment allows for correction through supplementation with vitamin D (25–50,000 IU per week for 4−8 weeks) and calcium (1000–1200 mg per day) [34,35]. Given the increased risk of osteoporosis in patients over the age of 65, screening for serum vitamin D levels appears to be particularly important in this population. Different types of screening were included in a multidisciplinary prehabilitation model by Di Leone [19], which focuses mostly on lowering distress among oncological patients. In the nutritional segment, the authors implemented anthropometric measurements and body composition analysis using segmental multi-frequency bioelectrical impedance analysis and nutritional screening (before the beginning and during therapy), in accordance with World Cancer Research Fund International (WCRF) recommendations [36], in order to lessen the toxicity of therapy and improve patients’ outcomes including QOL. The importance of nutritional screening for better treatment tolerance and higher QOL, before the start of oncological treatment, is also noticed by Shaw [21]. After screening, the aspects to consider while adjusting an optimal dietary strategy are nutritional impact symptoms, secondary diseases, type of planned therapy, and its toxicity or performance status. Accounting for these may improve management of nutritional impact symptoms and help to address patients’ nutritional needs precisely. The author also refers to articles [37,38] describing the need to implement nutritional support in the pre-cachexia stage, but the optimal time has not been specified yet. Collectively, these factors highlight that every oncology patient, including those awaiting chemotherapy, should undergo anthropometric assessment and body composition analysis to evaluate nutritional status and the risk of cachexia. Subsequently, a qualified dietitian should provide individualized nutritional recommendations aimed at minimizing chemotherapy’s side effects and reducing its toxicity.

### 3.2. Physical Activity Interventions

In prehabilitation models, either aerobic training, resistance training, or a combination of both is commonly implemented. Only one randomized clinical trial [39] conducted prior to chemotherapy was found in Mina [13] and Wagoner’s [22] reviews, performed among patients with BC (*n* = 24). Kirkham [39] aimed to determine the influence of a short but intense session of aerobic exercise 24 h prior to the first dose of doxorubicin. The session consisted of a 10-min warm-up, followed by 30 min of vigorous exercise (at 70% of the individual’s maximum heart rate reserve) and ending with a 5-min cool-down. According to the results, 24 h after the intervention, participants’ cardiac function showed less decline than in the control group. The researchers observed reduced acute release of N-terminal pro-B-type natriuretic peptide, a biomarker used to assess the severity of heart failure after the first dose of the cytostatic drug. This is a highly promising finding, suggesting that even short bouts of physical activity may have a significant impact on cardiotoxicity—a common adverse effect of chemotherapy. However, the sample size was very small, and further experimental studies are needed to draw more robust conclusions. Another common chemotherapy-related complication, apart from cardiotoxicity, is neurotoxicity, occurring in up to 75% of women with BC [40,41]. Its symptoms, along with co-occurring comorbidities, reduce patients’ QOL and may lead to treatment discontinuation. To investigate its potential prevention, Gonzalez-Santos [20] published an ATENTO (Adjusting the Dose of Therapeutic Exercise to Prevent Neurotoxicity Due to Anticancer Treatment) protocol for a clinical trial involving female breast cancer patients undergoing adjuvant chemotherapy. Two groups of participants receiving chemotherapy based on anthracycline or taxane regimens were defined. The ATENTO-A group includes patients prior to treatment, while the ATENTO-T group consists of those already undergoing treatment. The intervention is planned to last eight weeks and is multimodal. Each training session consists of three components: aerobic exercises, strength training, and vagal activation techniques (VATs) (see Table 4).

The program also promotes home-based physical activity, as defined by Tudor-Locke and Bassett [42], involving an increase in the number of steps taken throughout the day. Patients will be supervised by professionals in therapeutic exercise (TE), and parameters such as strength, total steps, and exercise intensity will be monitored using FITBIT activity bracelets, a mobile app (ATOPE+), and heart rate tracking. Assessments will take place at four time points: 2−4 days after diagnosis, 1 or 2 weeks after adjuvant chemotherapy for BC, and at 6- and 12-month follow-ups. Data collection will be based on a variety of scales and tests, as presented in Table 5.

The ATENTO protocol is a highly promising one, featuring a comprehensive assessment; however, its outcomes remain unavailable as the experimental study has not yet commenced. The physical activity component is also included in Di Leone’s multidisciplinary prehabilitation model [19]. As with nutritional prehabilitation, the model incorporated the WCRF International recommendations regarding physical activity [36] to be applied prior to chemotherapy, aiming to improve patients’ QOL, optimize outcomes, and lower therapy-related toxicity. In line with WCRF International guidelines, Wagoner [22] highlights the significance of exercising in the prehabilitation phase, referring to Campbell [53] and Schmitz [54], who emphasizes its role in mitigating treatment-related side effects such as fatigue and reduced physical function. For certain patients—such as those with lung cancer whose initiation of treatment is uncertain (the so-called ‘orange group’ according to Phillip’s concept [27])—incorporating exercise into the prehabilitation period may serve as a gateway to therapy by improving patient functional capacity and thereby increasing treatment rates. For patients classified within the ‘green group’, the author assumes that physical activity may serve as the foundation of prehabilitation interventions. Starting physical activity before chemotherapy seems to help reduce side effects, support better response to treatment, and improve patients’ QOL.

### 3.3. Counselling Interventions

To enhance adjustment to cancer and promote adherence to oncological treatments, Di Leone [19] incorporated a psychological intervention into her multidisciplinary model, based on the National Comprehensive Cancer Network (NCCN) Clinical Practice Guidelines in Oncology [55]. The clinical tools employed included GSES (General Self-Efficacy Scale), the Distress Thermometer (DT), and the Hospital Anxiety and Depression Scale (HADS). These are well-established clinical tools that are easy to apply in clinical practice and capable of early identification of patients who particularly require psychological support, especially since not all healthcare centres routinely provide psychological support to every patient. The main goal of psychological intervention in Di Leone’s study was to assess the risk of treatment-related distress while determining both protective and dysfunctional psychosocial factors. Di Leone proposed psychological assessment based on an interview divided into four main parts: individual and family psychological history (socio-demographic information; family history of cancer), clinical observation and psychological state (psychological assessment; emotional response to diagnosis), psychobiological rhythms and lifestyle (eating behaviour; sleep quality), and protective and risk psychological factors (emotional resources; oncological distress factors). As part of the counselling process, emotional eating was specifically evaluated, and patients were offered support through psychoeducational groups. The necessity of regular psychological support for patients with lung cancer is also mentioned by Phillips [27].

### 3.4. Smoking Cessation

Alongside interventions targeting psychological support, adequate nutrition and appropriate physical activity, smoking cessation is also one of the most frequently recommended strategies to optimize patients’ baseline condition prior to treatment. It plays a crucial role in reducing the risk of developing secondary lung cancers—that is, new tumour sites emerging after treatment or during follow-up [27]. Mina’s review [13] emphasizes the critical role of quitting smoking post-diagnosis, since tobacco use can alter the pharmacokinetics of the prescribed drug therapy. Mina refers to conclusions drawn in the expert report from 2017 [56], which state that the most effective approach to smoking cessation appears to be the combination of behavioural interventions and pharmacologic therapy. It seems advisable that clinicians working with lung cancer patients emphasize smoking cessation and provide information about available therapies and their potential benefits for treatment outcomes. Despite the numerous advantages of smoking cessation after receiving a cancer diagnosis—including improved treatment efficacy, increased survival rates, and enhanced quality of life—up to 60% of patients continue to smoke [25]. This underscores the significance of the issue and indicates the need for greater emphasis on implementing optimized support for patients in this area. Cedzyńska [25] highlights that incorporating psychological counselling elements within the prehabilitation phase is essential. She recommends delivering a brief, personalized message that emphasizes the benefits of cessation, without inducing guilt. Ideally, patients should have access to a smoking cessation support centre affiliated with the oncology clinic. It is noted that the presence of an oncologist is not essential, as trained healthcare professionals (e.g., nurses and psychologists) are adequately equipped to provide such support.

### 3.5. Other Interventions

Since cachexia significantly affects survival, it may determine whether treatment can be initiated in patients with lung cancer. Consequently, Phillips [27] recommends considering anti-cachectic agents in this patient population. Flemming et al. [26], having found that insomnia affects 46% of newly diagnosed breast cancer patients and is associated with reduced overall survival irrespective of other prognostic factors, developed the Investigating the Value of Early Sleep Therapy trial protocol. The study aims to evaluate the feasibility and acceptability of sleep restriction therapy (SRT) in this cohort, alongside evaluations of insomnia severity, psychological outcomes, and circadian rest–activity rhythms. Fifty newly diagnosed patients with BC suffering from acute insomnia will be recruited prior to neoadjuvant chemotherapy, radiotherapy, or surgery and randomized to receive either SRT or sleep hygiene education. Over a four-week period, participants will attend scheduled educational sessions and receive nursing support. Data on the study’s outcomes are anticipated to become available by April 2026. This protocol highlights the importance of addressing insomnia early among breast cancer patients, as effective management of sleep disturbances may improve overall survival and quality of life during treatment.

## 4. Discussion

Our integrative review included ten studies; however, only a few provided detailed descriptions of pre-chemotherapy interventions, which are integral components of multidisciplinary protocols for future clinical research. In both Mina’s [13] and Wagoner’s [22] reviews, we identified randomized clinical trials (RCTs) [39] in which less declined cardiac function was found after implementing aerobic training 24 h prior to the first dose of doxorubicin, compared with the control group. Despite this promising finding, no other clinical trials were found. Most of the remaining evidence consisted of mixed intervention protocols, which have not been further tested or developed in subsequent research.

The limited number of rigorous studies and the lack of standardized intervention protocols underscore a significant gap in the literature concerning effective prehabilitation strategies tailored specifically for patients awaiting chemotherapy. This scarcity of high-quality data restricts the ability to draw robust conclusions regarding the efficacy of prehabilitation components, such as physical activity, nutritional support, psychological counselling, and smoking cessation, in improving clinical outcomes and QOL.

Furthermore, the available literature consists mainly of narrative reviews and study protocols, with no actual interventions implemented or outcome measures reported. This poses a major challenge for synthesizing evidence and applying findings to clinical practice. To advance this field, future research must focus on designing well-powered, randomized controlled trials with clearly defined, replicable intervention protocols, standardized outcome assessments, and long-term follow-up. Such studies are essential to establish both the clinical utility and cost-effectiveness of prehabilitation programmes in oncology.

Given the limited number of studies retrieved in our review, we also included a proposal for a multidisciplinary prehabilitation protocol in the Discussion Section as shown in Appendix A). This protocol integrates all the key aspects described in the reviewed studies, supported by findings from the wider prehabilitation literature, including studies in surgical oncology. Our aim is to provide a practical framework and potential guidelines for future researchers to develop and evaluate prehabilitation interventions before chemotherapy, ultimately improving QOL and treatment outcomes for patients undergoing such therapy.

The conclusion that sarcopenia screening is crucial is supported by a comprehensive systematic review [57], which demonstrates a strong association between sarcopenia in patients with cancer and poorer chemotherapy outcomes, including chemotherapy-related toxicity and reduced overall survival. This finding aligns with the observations of Jang [24]. Notably, patients with upper gastrointestinal, pancreatic, respiratory tract, and head and neck tumours are at highest risk of sarcopenia [57]. To prevent sarcopenia and support immune competence, sufficient protein intake is necessary to maintain lean body mass. The latter is essential in fighting tumorcells. Even though experts on ESPEN (European Society for Clinical Nutrition and Metabolism) guidelines [58] underline that oncological diseases themselves do not impair the synthesis of protein, its intake after cancer diagnosis should be over 1.0 g/kg of body mass and up to 1.5 g/kg of body mass. The conclusions of Morton’s systematic review [59] are in line with these recommendations: it showed that protein intakes of more than 1.6 g/kg of body mass did not further increase fat-free mass when combined with resistance exercise training. Conversely, another recent systematic review [60] suggests that intakes below 1.2 g/kg of body mass are associated with muscle deterioration, whereas a dose of 1.4 g/kg of body mass supports muscle preservation. In Mina’s review [13], the authors cite ESPEN guidelines recommending protein intakes of 1.2–2.0 g/kg of body mass. However, the dosage specified in these guidelines refers to enteral nutrition, whilst our protocol and previously mentioned papers concern oral feeding. ESPEN [30] does not specify a superior type of protein based on bioavailability, although whey is favoured over casein due to higher concentration of leucine. It is acknowledged as primary amino acid impacting muscle protein synthesis (MPS). However, it should be noted that the optimal anabolic effect on muscles is achieved when adequate protein intake is combined with the implementation of resistance training [61]. Moreover, distributing protein evenly across meals may enhance 24-h protein anabolic response, especially in older adults [62]. Therefore, it is fundamental to understand the importance of preventing muscle loss, as this process is an integral part of cachexia. During neoadjuvant treatments, weight loss may differ between 4 and 12 kg, and majority of this loss is muscle mass. Alongside a protein-rich diet, strategies to mitigate nausea, improve appetite and manage pain may further support nutritional intake [63].

Omega-3 polyunsaturated fatty acids are one of the key pharmaconutrients in the context of oncological treatments. They decrease inflammation caused by cancertumors, which may support maintaining body weight, increased appetite, or higher food intake. Although, at the time of writing this review, no trials on omega-3 supplementation in oncology patients prior to chemotherapy have been identified—and none of the retrieved studies mentioned this nutrient—they indisputably hold potential in prehabilitation settings. The ESPEN guidelines [58] highlight no major safety concerns, despite weakened compliance in patients experiencing a fishy aftertaste or belching. A weak recommendation for the use of long-chain n-3 fatty acids has been established. This is supported by several studies, suggesting their potential in neurotoxicity protection linked to many chemotherapeutics, reducing muscle proteolysis, enhancing MPS, increasing or maintaining appetite as well as food intake and body weight, and contributing to better QOL [30,58]. Regarding optimal daily intake during prehabilitation, our recommendations are based on ERAS guidelines [30]. The most commonly used dose in the cited research reporting benefits was 2.2 g of the EPA (eicosapentaenoic acid) a day. A recent review evaluating the importance of the EPA-to-DHA (docosahexaenoic acid) ratio in oncological therapy [64] suggests that a ratio below 2—or even lower than 1—may be most beneficial. Thus, future research on the topic should focus on fish oil supplements containing a 1:2 DHA-to-EPA ratio or a higher proportion of DHA.

There is growing interest among researchers in the potential role of vitamins in mitigating chemotherapy-related side effects, such as oral and intestinal mucositis [65]. Nevertheless, among the records reviewed, only Bertrand [23] emphasized the importance of adequate serum vitamin D levels in preventing bone loss among older patients undergoing chemotherapy. However, no specific recommendation regarding the use of this vitamin in oncological patients has been established in the literature. According to the ESPEN guidelines [30], vitamin D supplementation does not decrease the risk of fractures by more than 15%, and there is a lack of data regarding patients with deficient vitamin D concentrations. In consideration of this, we conclude that correcting serum vitamin D levels may be beneficial for patients undergoing chemotherapy, and therefore, screening appears to be warranted. In a recent systematic review [65] focused on the role of vitamins in treating intestinal mucositis, we identified one human trial involving vitamin D [66]. The authors concluded that its supplementation in patients with a deficiency may lower the incidence of chemotherapy-induced diarrhea. In an insightful review on breast cancer patients receiving NAC (*n* = 1291) [67], in which circulating serum vitamin D levels were measured, a strong association was found between pre-treatment vitamin D deficiency and increased odds of non-pathological complete response as well as overall reduced treatment efficacy. Another human study suggested that vitamin D insufficiency may increase the risk of chemotherapy-induced peripheral neuropathy (CIPN) in response to paclitaxel treatment [68]. In view of these findings, we recommend correcting serum vitamin D levels in all patients scheduled for chemotherapy if concentrations fall below 30 ng/mL.

Vitamin E is another nutrient of interest in the context of prehabilitation. CIPN is a major side effect of drug therapy, affecting between 18% and 85% of patients. The use of vitamin E to prevent this condition is disputable, although a 2021 meta-analysis of randomized trials [69] showed that among patients treated with cisplatin, the incidence of peripheral neuropathy decreased and neurotoxicity scores were lower when 600 mg/day of vitamin E was administered. No such effect was observed in patients treated with paclitaxel. Since the supplementation period lasted three months, it is difficult to classify vitamin E as a component of prehabilitation. More trials concerning these pharmaconutrients over shorter periods of time are necessary. An interesting meta-analysis [70] found that the use of vitamin A and E after a breast cancer diagnosis, did not influence overall survival, whereas vitamin C supplementation significantly decreased total deaths. The lack of efficacy for vitamin A supplementation was also confirmed in a recent systematic review [71]. ESPEN guidelines [58] recommend that “vitamins be supplied in amounts approximately equal to the recommended daily allowance”.

Physical activity during TDT may not only prevent muscle tissue loss but also mitigate some of the adverse effects of chemotherapy. Cardiovascular disease remains the leading cause of morbidity and mortality among individuals who have undergone cancer treatment [72], highlighting the importance of incorporating physical activity into prehabilitation protocols. Kirkham [39], in an RCT, reported a reduction in cardiotoxicity resulting from physical activity undertaken prior to doxorubicin treatment. However, it should be noted that the sample size was small (*n* = 24), and, to date, no other human RCTs have supported these findings. In preclinical research, Wonders [73] demonstrated preserved systolic function in rats subjected to a single bout of high-intensity aerobic training prior to chemotherapy, while the control group exhibited a 45% decrease in function. Similarly, another rat study [74] showed that 12 weeks of aerobic training prior to doxorubicin therapy significantly preserved cardiac function. Five days post-treatment left ventricular function in the trained group was over 40% higher compared to sedentary controls. Moreover, the mortality rate was 13% lower in the trained group. These findings underscore that, beyond the well-established benefits of regular aerobic exercise, even a single session of physical activity before chemotherapy may play a critical role in preserving cardiac function. While there is a paucity of literature on physical activity prior to chemotherapy, a recently published comprehensive systematic review and meta-analysis [75] evaluated the increase in VO_2_max in oncological populations depending on the therapeutic physical exercise (TPE) programme implemented. TPE is most effective when supervised—either in person or online—by a healthcare professional. VO_2_max is a parameter used to measure cardiorespiratory fitness (CRF), and individuals with higher CRF levels show a significantly decreased risk of mortality. The conclusions of the review suggest that, among men, the most effective intervention consisted of moderate-to-vigorous physical activity performed for a minimum of 35 min, three times per week over a 12-week period. High-intensity interval training and aerobic exercises were the most effective. Following the intervention, improvements in VO_2_max ranged from 1.5% to 4.2%. In women, VO_2_max improved by 0.3% to 53%. Similarly, the best results were observed following a 12-week intervention consisting of aerobic and resistance training, performed three times per week at moderate-to-vigorous intensity. To assess VO_2_peak, Gonzalez-Santos [20] used a cardiopulmonary exercise test with a gas analyzerbased on the University of Northern Colorado Cancer Rehabilitation Institute (UNCCRI) protocol [51]. The findings of this study support the use of the UNCCRI treadmill protocol as a valid, cancer survivor-specific method for assessing VO_2_peak. The high correlation between directly measured and estimated VO_2_peak values confirms the accuracy of American College of Sports Medicine metabolic equations, aligning with results observed in established protocols such as the modified Balke and Naughton tests. Given its tailored intensity progression and safety profile, the UNCCRI protocol offers a reliable alternative for VO_2_peak assessment in cancer populations. Next to cardiotoxicity and the decrease in VO_2_max, cancer-related fatigue is one of the primary symptoms associated with cancer and has been the focus of numerous studies during chemotherapy. However, a recent American meta-analysis [24] found no significant reduction in cancer-related fatigue resulting from the implementation of aerobic and/or resistance training compared to usual care. This highlights a gap in the literature and underscores the need for further research on physical interventions prior to the initiation of chemotherapy. Physical exercises, whether standalone or as part of multimodal interventions, are widely studied in preoperative care. Although the aims of prehabilitation differ between patients treated with chemotherapy and those undergoing surgery, findings from surgical prehabilitation studies may offer valuable insights for chemotherapy-related research.

Receiving a cancer diagnosis is often accompanied by fear and emotional distress, and in severe cases may lead to anxiety, depression, or even post-traumatic stress disorder (PTSD) [76]. Counselling is a key component of a comprehensive prehabilitation framework, as highlighted in the works of Mina [13], Wagoner [22], and Di Leone [19]. Mina emphasizes the importance of distress screening in order to tailor interventions to the most vulnerable patients. While distress screening is a standard practice in hospitals in the United States, it still needs to be more widely established across Europe. Several diagnostic tools have been proposed by Yi [76], who recommends the Distress Thermometer as a first-line screening method. However, he underscores the need to use an additional assessment tool when a score of 4 or higher is obtained. In addition to the DT, Di Leone also recommends the use of the Hospital Anxiety and Depression Scale (HADS), aligning with Yi’s approach. Given the variation in screening practices across Europe, a simple two-step approach—using the DT followed by HADS for patients with higher scores—could offer a practical and easy-to-implement solution. Future research should focus on how such screening processes can be integrated into routine prehabilitation care. Mina also reports the positive effects of interventions such as yoga, stress management training, coping strategies, muscle and breathing relaxation techniques, and meditation on mood, anxiety, depression, and even postoperative fatigue. Similarly, Wagoner [77] discusses the beneficial impact of physical activity, particularly Pilates, on reducing depressive symptoms. He also cites a 2019 systematic review [78] that supports these findings, demonstrating that aerobic exercise (primarily walking) improved QOL in breast cancer survivors, reduced depressive symptoms, and enhanced sleep quality. In 2022, a comprehensive review by Grimmett [79] summarized the existing literature on psychological support for patients following a cancer diagnosis. The authors highlighted strong evidence supporting the positive effects of physical exercise on mental health outcomes. Psychological support provided by a specially trained specialist also yields measurable benefits. In the review cited by Grimmett, a unimodal prehabilitation model was evaluated [80], in which patients awaiting surgery or neo-/adjuvant treatment who received psychotherapeutic support prior to treatment exhibited significantly lower levels of depression, distress, and anxiety. Additionally, those patients demonstrated better immune function and improved physical condition compared to a control group that received only standard care. Introducing psychological support and physical activity in the pre-treatment phase may significantly reduce emotional distress in cancer patients. However, there remains a lack of research specifically focusing on patients prior to chemotherapy, highlighting the need for further studies to better understand their potential impact on the course of this particular treatment.

Smoking cessation, alongside nutritional and physical interventions, is one of the main components of comprehensive prehabilitation and is widely used prior to surgical treatment [81]. While smoking is recognized as a risk factor for postoperative complications, it may also affect chemotherapy by altering pharmacokinetics, as noted in the study by Mina [13]. In a 2018 literature review [82], the authors cautiously concluded that nicotine may weaken the effectiveness of anticancer drugs such as cisplatin, paclitaxel, and gefitinib by activating nicotinic acetylcholine receptors. This activation may lead to increased cancer cell survival and reduced susceptibility to apoptosis; however, these assumptions are based on in vitro studies. Their own in vivo study did not show a significant impact on the efficacy of paclitaxel in lung cancer patients. The conclusions of another study [83] focusing on the response to neoadjuvant treatment showed that former smokers had a better response to chemotherapy compared to active smokers. Similarly, patients with pancreatic cancer also had lower survival rates than those who quit after diagnosis [84]. In Mina’s paper [13], the author refers to a study [85] in which researchers found that behavioural therapy combined with pharmacological treatment was the most effective method for smoking cessation in patients. The methods used in the study included nicotine replacement therapy and motivational interviewing. “Apart from disease site and stage, abstinence from smoking is considered the strongest predictor of survival in cancer patients who have ever smoked”, as stated by Cedzyńska [25] in her publication. She recommends delivering a patient-tailored message on the importance of smoking cessation, highlighting the potential treatment-related benefits. Taken together, although further research is needed, these data indicate that nicotine may modulate the response to chemotherapy and influence survival rates. Therefore, smoking cessation should be considered an essential component when developing a prehabilitation protocol. 

Sleep disturbance is a crucial aspect of oncological care, particularly among patients with BC, as examined in the study by Fleming [26]. The application of sleep restriction therapy during chemotherapy represents the first trial of its kind; however, no definitive conclusions can be drawn yet, as the trial is still ongoing. Notably, sleep restriction therapy is also recommended in the American Academy of Sleep Medicine’s clinical practice guideline [86] as an option for clinicians to consider. Cognitive behavioural therapy has been implemented in studies on insomnia among oncology patients, resulting in significant improvements in sleep quality both during and after treatment [87,88]. However, the practicality of such interventions remains questionable—behavioral therapy programs typically last around one to two months, whereas neoadjuvant chemotherapy for breast cancer may begin as early as two weeks post-diagnosis. A pilot study by Zhang [89] involving patients with BC employed 15 acupuncture sessions. In the intervention group, 56.5% of participants discontinued the use of sleeping medications, compared to 14.3% in the control group (P = 0.011). Although these results are promising, the duration and high cost of the intervention may limit its feasibility in routine clinical practice.

### Limitations

Before delving deeper into the interpretation of our findings, it is essential to acknowledge several key limitations. Firstly, the limited number of eligible studies narrowed the scope of our analysis and constrained our ability to draw definitive conclusions. Secondly, although the analysis was conducted across multiple databases, only four keywords were used in the search. Other researchers may have used different terminology in their publications, making such studies difficult to identify using our approach. Thirdly, although all included articles were peer-reviewed, the heterogeneity of study methods complicated the synthesis of findings. Fourthly, we did not assess methodological quality or risk of bias. Fifthly, studies involving patients receiving nutrition via non-oral routes (e.g., nasogastric or nasojejunal tubes, feeding ostomies, or parenteral nutrition) were excluded, even though this patient group requires special consideration when preparing for planned treatment. Sixthly, the proposed interventions in our protocol are partly based on recommendations derived from the available literature rather than on solid evidence or official guidelines issued by professional societies. Moreover, the proposed strategies are general in nature and not tailored to specific patient groups or particular types of cancer, reflecting the limited scope of literature currently available.

Additional limitations arise from the studies included in this review. The study designs are heterogeneous, employing diverse research methods and reflecting varying perspectives on prehabilitation. Patient populations may also be inadequately represented. Notably, six out of the ten evaluated papers focused exclusively on breast cancer, limiting the generalizability of findings to other cancer types. Furthermore, data concerning elderly patients remain scarce, which makes it difficult to extrapolate findings to older individuals, who often face unique treatment challenges related to comorbidities and geriatric syndromes. A major limitation is the paucity of experimental studies, preventing the formulation of strong quantitative conclusions. These limitations underscore the need for larger, well-controlled, and standardized studies to build a more robust evidence base for optimizing prehabilitation strategies in cancer care.

Despite these limitations, our study provides valuable insights into the current state of knowledge in this field. To the best of our knowledge, this is the first review to comprehensively integrate published studies on this topic. All included studies are peer-reviewed and were published between 2017 and 2024, making them fairly recent. By synthesizing studies of various types from multiple databases, we were able to identify consistent patterns and gain a deeper understanding of the critical elements shaping this area of research. Furthermore, our review may serve as a cornerstone for future research, which is urgently needed to improve both survival rates and the quality of life in patients undergoing chemotherapy.

## 5. Conclusions

Prehabilitation prior to chemotherapy has the potential to improve treatment tolerance, clinical outcomes, and patient well-being, and should become a routine element of oncological care. There is limited research available on this topic and no official recommendations exist for this patient group, although our review highlights several promising interventions.

Physical activity—particularly resistance training—may help preserve muscle strength and improve cardiorespiratory fitness. Notably, a study by Kirkham et al. suggests that even a single aerobic session may reduce doxorubicin-induced cardiotoxicity, which shows promise and should be further tested in clinical settings. Psychological distress is common among oncology patients; thus, routine distress screening and timely access to psychosocial support should be prioritized. Research should explore optimal screening tools and intervention timing.

Smoking cessation remains essential, particularly in lung cancer patients. Future studies should evaluate how integrated cessation programmes within prehabilitation frameworks affect treatment outcomes.

Adequate protein intake and sarcopenia screening should be standard from diagnosis onward to preserve muscle mass. Nutritional interventions—including omega-3 fatty acids and vitamin D correction—show potential but require more robust clinical evaluation.

To advance this field, future high-quality randomized trials should focus on single-component interventions to clarify their feasibility, mechanisms of action, and translatability across cancer populations. Our review provides a foundation for designing such studies and integrating evidence-based prehabilitation into routine clinical practice.

## Figures and Tables

**Figure 1 cancers-17-02670-f001:**
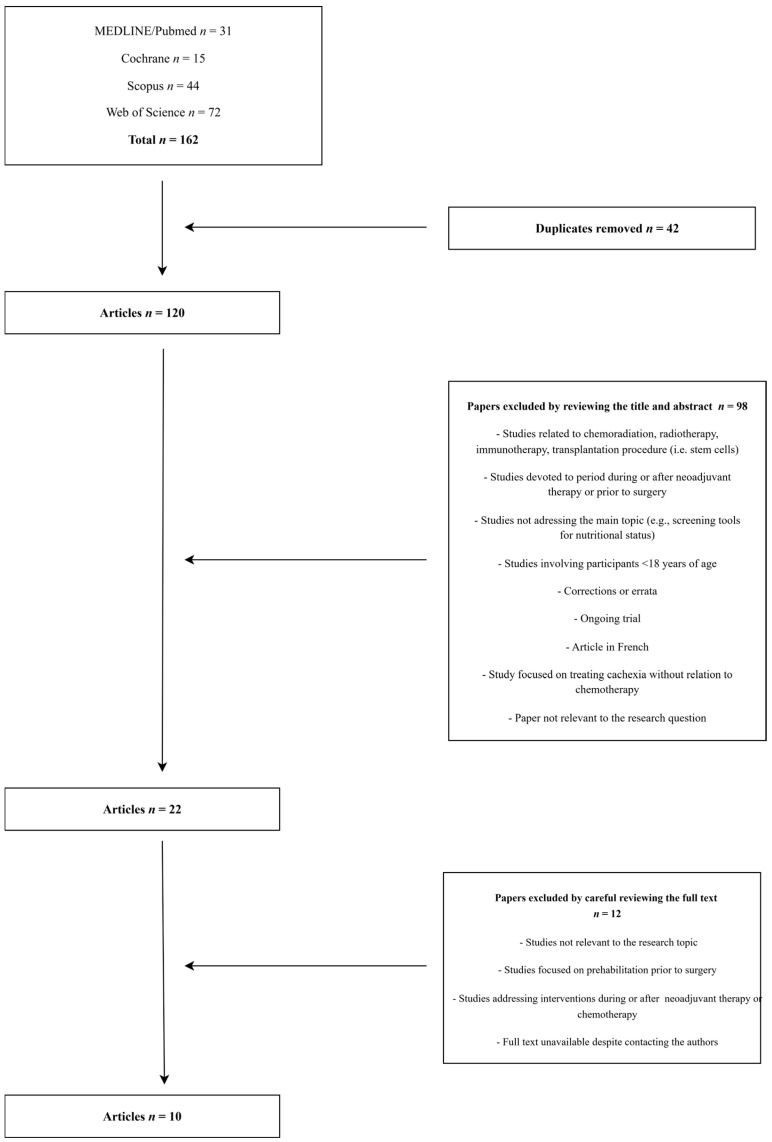
Diagram of screening strategy.

**Figure 2 cancers-17-02670-f002:**
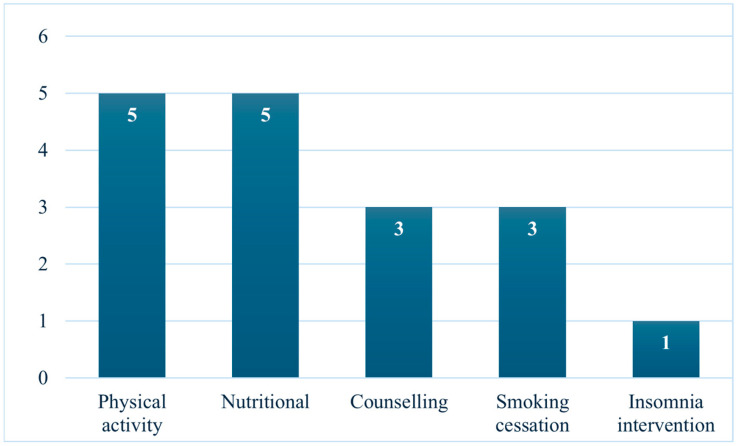
Frequency of prehabilitation intervention types in the reviewed studies.

**Table 1 cancers-17-02670-t001:** Searching procedure.

Database Name (Filters Applied)	Search Query Used	Results (*n*)
MEDLINE/PubMed (Title/Abstract)	((prehabilitation) AND (chemotherapy OR neoadjuvant OR “drug therapy”)) NOT (surg*)	31
Cochrane (Title Abstract Keywords)	((prehabilitation) AND (chemotherapy OR neoadjuvant OR “drug therapy”)) NOT (surg*)	15
Scopus (Article Title, Abstract, Keywords)	( TITLE-ABS-KEY ( prehabilitation ) AND TITLE-ABS-KEY ( chemotherapy OR “drug therapy: OR neoadjuvant ) AND NOT TITLE-ABS-KEY ( surgery ) ) AND PUBYEAR > 2009 AND PUBYEAR < 2026 AND ( LIMIT-TO ( LANGUAGE , “English” ) )	44
Web of Science (Topic)	(TS=(Prehabilitation) AND TS=(chemotherapy OR “drug therapy” OR neoadjuvant)) NOT TS=(surgery)	72
Total		162

MEDLINE/PubMed—Medical Literature Analysis and Retrieval System Online/National Library of Medicine; * the phrase “drug therapy” would have been more precise, but at the time of the search, the broader Boolean was applied.

**Table 2 cancers-17-02670-t002:** Inclusion and exclusion criteria.

Inclusion Criteria	Exclusion Criteria
Human subjects (≥18 years old)	Studies concerning radiotherapy, chemoradiotherapy, surgery, hormonotherapy, immunotherapy
Sampling = patients prior to chemotherapy	Studies concerning ERP and ERAS
Peer reviewed papers	Studies describing interventions during or after the treatment
Papers published in English	
Published between 2010 and January 13th 2025	

ERP—Enhanced Recovery Protocol; ERAS—Enhanced Recovery After Surgery.

**Table 3 cancers-17-02670-t003:** Characteristics of included studies.

Reference	First Author’s Name, Journal, Year of Publication	Digital Object Identifier	Design of the Study	Type of Prehabilitation	Type of Cancer
[13]	Mina, PM&R, 2017	10.1016/j.pmrj.2017.08.402	Narrative review	Physical activity	Breast
				Nutrition	
				Counselling	
				Smoking cessation	
[19]	Di Leone, J. Pers. Med., 2021	10.3390/jpm11050324	Narrative review	Physical activity	Breast
				Nutrition	
				Counselling	
[20]	González-Santos, Res Nurs Health, 2021	10.1002/nur.22136	Study protocol for a randomized controlled trial	Physical activity	Breast
[21]	Shaw, Proc. Nutr. Soc., 2021	10.1017/S0029665120007041	Narrative review	Nutrition	Gastrointestinal
[22]	Wagoner, Curr Oncol, 2022	10.3390/curroncol29070383	Narrative review	Physical activity	Breast
[23]	Bertrand, Joint Bones Spine, 2023	10.1016/j.jbspin.2023.105549	Narrative review	Nutrition	Non-specified
[24]	Jang, Support Care Cancer, 2024	10.1007/s00520-024-08532-0	Systematic review and meta-analysis	Nutrition	Breast
[25]	Cedzyńska, Nowotwory J Oncol, 2024	10.5603/njo.101552	Narrative review	Smoking cessation	Non-specified
[26]	Fleming, PLoS ONE, 2024	10.1371/journal.pone.0305304	Pilot randomized controlled trial protocol	Insomnia intervention	Breast
[27]	Phillips, Curr Opin Support Palliat Care, 2024	10.1097/SPC.0000000000000716	Narrative review	Physical activity	Lung

**Table 4 cancers-17-02670-t004:** ATENTO session in details.

Session’s Parts (in Order)	Time of Duration	Sport Equipment Used
Warm-up	8–10 min	Elliptical trainer, dumbbells, elastic bands, mat
2. Cardiovascular and strength exercises	Total of 60 min (cardiovascular 10–30 min, strength 30–50 min)	
3. VATs	20 min	

VATs—vagal activation techniques.

**Table 5 cancers-17-02670-t005:** Assessment of patients with breast cancer.

Parameters Assessed	Assessment Tool Used
Quality of life (QOL)	European Organization for Research and Treatment of Cancer Quality of Life Questionnaire Core 30 (EORTC QLQC30) version 3.0 [43]
Mental flexibility, speed of processing, and executive function	The Trail Making Test (TMT)
Memory and cognitive processing speed	Wechsler Adult Intelligence Scale (Wechsler, 2008) [44]
Neuropathic symptoms	The EORTC QLQ-Chemotherapy-induced peripheral neuropathy (QLQCIPN20) (Postma, 2005) [45]
Peripheral sensory neuropathy	Semmes–Weinstein monofilaments (SWMs) (Bell-Krotoski, 1995) [46]
Anxiety and depression	The Hospital Anxiety and Depression Scale (HADS) (Zigmond & Snalth, 1983) [47]
Cancer-related fatigue	The Piper Fatigue Scale-Revised (PFS-R) (Piper, 1998) [48]
Pain severity and the interference of pain with daily activities	Brief Pain Inventory (BPI) (Poquet & Lin, 2016) [49]
Exploring the quadriceps, deltoid, trapezius, and cervical muscles bilaterally	Pressure pain thresholds and algometer
Sleep quality	The Pittsburgh Sleep Quality Index (PSQI) (Buysse, 1989) [50]
VO_2_peak	Cardiopulmonary exercise test (Medisoft, 870A treadmill) and Jaeger MasterScreen® CPX gas analyser (protocol of the University of Northern Colorado Cancer Rehabilitation Institute) [51]
Whole-body balance	Flamingo test
Physical activity or inactivity level	International Physical Activity Questionnaire (IPAQ) (Craig, 2003) [52]
Estimates lean body mass, fat mass, abdominal adipose tissue, and body max index (kg/m^2^)	InBody 720 impedance meter

VO_2_peak—peak oxygen uptake.

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
