# Peer review of "Prehabilitation Prior to Chemotherapy in Humans: A Review of Current Evidence and Future Directions"

_cancers, 2025, doi:10.3390/cancers17162670_

Round 1
Reviewer 1 Report
Comments and Suggestions for Authors
This is a well-written review. The authors clearly spent much time on this paper, and there is a need for this kind of assessment in the literature.
I have the following comments:
- The abstract results should be rewritten. It focuses on one small trial and ignores the rest of the literature. It should list the number of studies reviewed (n=10). Further, most of these were reviews of breast cancer. This indicates, at least to me, that the issue of prehabilitation knowledge varies by cancer type. For example, there were no previous reviews focusing on prostate cancer.
- Line 42 abstract. Please specify that you are referring to adjuvant chemotherapy and not neoadjuvant?
- I had to read this article twice before fully understanding it. The methods state the search is based on prehabilitation, but the keywords indicate any chemotherapy. It's not entirely clear why this is upfront. One has to finish the article to understand this. For example, line 178 refers to the Mina study. But that study as well as other cited refers to any chemotherapy and not to neoadjuvant therapy. The reader is left with the impression that the Mina and other findings are based on neoadjuvant literature.
- It seems the authors are making the case that treatment therapies thought to be helpful for chemotherapy are therefore relevant or should be studied for neoadjuvant therapy? This needs to be made more explicit in the introduction, if that is the case. But that isn't necessarily true. For example, increased protein intake is a treatment for cancer cachexia, which often occurs for advanced cancer when symptoms are severe. So, for example, increasing protein intake for neoadjuvant therapy for early-stage breast cancer doesn't seem relevant. The authors need to make the case. Further they state that the purpose of the review is make recommendations for clinical trials in neoadjuvant patients. Using their own example then, if protein intake is important for certain chemotherapy outcomes, is this sufficient to recommend for neoadjuvant patients as well?
- Along similar lines, the first paragraph of the conclusion cites evidence from surgically treated patients, but the current paper is based on patients without surgery receiving pretreatment. If the paper is simply making a comparison to note that the neoadjuvant literature is lacking that is okay. But then, again, if the idea is to use chemotherapy findings to develop clinical trials for nonsurgical adjuvant therapy, there are many dissimilarities between the groups (age, surgery, stage etc.).
Reviewer 2 Report
Comments and Suggestions for Authors
Overall Comments
Thank you for the opportunity to review your manuscript on this vital topic. I appreciate the work that you have undertaken to review this topic. I provide the following suggestions/ recommendations to enhance the paper. The comments primarily relate to the Materials and Methods section, which requires clarification and some minor wording adjustments. Your review has contributed to the knowledge of prehabilitation and will influence changes in practice.
Abstract:
Please consider rewording the first sentence into two, one that focuses on the patient and the other on the medical staff. The reason is that the current format is challenging to gain what you are endeavouring to state. Ask yourself, is it the patient complications or the medical staff complications?
Current format
“Chemotherapy presents challenges not only for the patient - by causing unpleasant side effects - but also for the medical staff, who must sometimes react promptly to complications to help the patient complete treatment in the best possible condition.”
Suggestions
Reword into two sentences or state
Medical staff must respond promptly to assist patients in completing their chemotherapy treatment if complications arise.
Suggestion
Abstract
Line 15: Change 'trained healthcare professionals' to 'registered' or 'educated' to meet international standards and conventions.
Background
Suggestion: Change line 28 from “available studies to published peer-reviewed studies. As available does not indicate that you sourced them from peer-reviewed journals.
Method section
I suggest stating in the methods section that your integrative review was guided by Torraco (2026), as it is not until Line 116 that you refer to the format that guided the integrative review. The reader will want to identify the method earlier.
Torraco, R.J. Writing Integrative Literature Reviews. Human Resource Development Review 2016, 15, 404-428, 686doi:10.1177/1534484316671606.
Materials section
In line 132, were the records imported from the research articles? If they were research articles, I suggest rewording 'records' to 'articles'.
In line 138, I suggest changing the start to state 'Extract research articles' instead of just 'research'.
Is Figure 1. Search strategy: a PRISMA Screening or a Flowchart? The title of Figure 1 needs to be changed, as it is not the search strategy; it is a screening where you state when research articles were excluded and for what reason.
I recommend that you expand the Materials ad Methods section to include how, who and what screening tool you used to review the research papers. Consider the following:
- Did you use a screening tool, i.e. an Appraisal Checklist or a Screening tool? If so, please include the name of the tool and provide a reference.
- Did all authors review each research article? Or did two authors review all manuscripts or some other number? What occurs if there is a disagreement in the review process between reviewers? Did a third reviewer become involved?
Results
You have provided a good discussion of the results of your integrative review and highlighted the critical areas for consideration in establishing prehabilitation guidelines.
Discussion
Your discussion is appropriate and informative for the reader to decide the value of your manuscript and its contribution to the development of guidelines.
Limitations
They are well described and discussed, and the importance is noted.
Conclusion
I suggest condensing the conclusion to highlight the critical key points that clearly demonstrate how your review contributes to the field of knowledge and clinical practice.
